

# Identification and expression profiles of candidate chemosensory receptors in *Histia rhodope* (Lepidoptera: Zygaenidae)

Haibo Yang, Junfeng Dong, Ya-Lan Sun, Zhenjie Hu, Qi-Hui Lyu and Dingxu Li

College of Forestry, Henan University of Science and Technology, Luoyang, Henan, China

## ABSTRACT

Insect olfaction and vision play important roles in survival and reproduction. Diurnal butterflies mainly rely on visual cues whereas nocturnal moths rely on olfactory signals to locate external resources. *Histia rhodope* Cramer (Lepidoptera: Zygaenidae) is an important pest of the landscape tree *Bischofia polycarpa* in China and other Southeast Asian regions. As a diurnal moth, *H. rhodope* represents a suitable model for studying the evolutionary shift from olfactory to visual communication. However, only a few chemosensory soluble proteins have been characterized and information on *H. rhodope* chemoreceptor genes is currently lacking. In this study, we identified 45 odorant receptors (ORs), nine ionotropic receptors (IRs), eight gustatory receptors (GRs) and two sensory neuron membrane proteins (SNMPs) from our previously acquired *H. rhodope* antennal transcriptomic data. The number of chemoreceptors of *H. rhodope* was less compared with that found in many nocturnal moths. Some specific chemoreceptors such as OR co-receptor (ORco), ionotropic receptors co-receptor, $CO_2$ receptors, sugar receptors and bitter receptors were predicted by phylogenetic analysis. Notably, two candidate pheromone receptors (PRs) were identified within a novel PR lineage. qRT-PCR results showed that almost all tested genes (22/24) were predominantly expressed in antennae, indicating that they may be important in olfactory function. Among these antennae-enriched genes, six ORs, five IRs and two GRs displayed female-biased expression, while two ORs displayed male-biased expression. Additionally, HrhoIR75q.2 and HrhoGR67 were more highly expressed in heads and legs. This study enriches the olfactory gene inventory of *H. rhodope* and provides the foundation for further research of the chemoreception mechanism in diurnal moths.

# INTRODUCTION

*Histia rhodope* Cramer (Lepidoptera: Zygaenidae), a diurnal moth, is a destructive forest pest widely distributed in China and other Southeast Asian regions (*Huang, 1980*). Although *H. rhodope* is an oligophagous pest, its larvae mainly feed on the leaves of *Bischofia polycarpa* (Chinese bishopwood), resulting in severe defoliation. Mature larvae always fall from the tree by spinning silk, which not only affects the appearance but also disturbs

Corresponding authors
Haibo Yang, hbyang@haust.edu.cn
Junfeng Dong, junfeng-dong@haust.edu.cn

human activities (*Huang, 1980*; *Yang et al., 2019*). Upon eclosion, males court females through nuptial flight in the afternoon. In addition, females release sex pheromones for attracting males. After mating, females then search for suitable oviposition sites and lay eggs (*Yang et al., 2019*). Therefore, the detection of sex pheromones is crucially important for reproduction.

Olfaction plays an important role in many behaviors in insects. Insects depend on their sensitive olfactory system to detect semiochemicals in complex environments (*Gadenne, Barrozo & Anton, 2016*; *Haverkamp, Hansson & Knaden, 2018*). A myriad of proteins is implicated in the process of olfactory perception. Chemoreceptors are key proteins that interact directly with odorants and play an important role in the specificity and sensitivity of insect peripheral chemoreception (*Leal, 2013*). Chemosensory receptors include odorant receptors (ORs) (*Clyne et al., 1999*), gustatory receptors (GRs) (*Agnihotri, Roy & Joshi, 2016*) and ionotropic receptors (IRs) (*Benton et al., 2009*). Additionally, sensory neuron membrane proteins (SNMPs), located in different dendrites of the olfactory receptor neurons, function in odor detection (*Rogers, Krieger & Vogt, 2001*).

Insect ORs were first identified in the *Drosophila melanogaster* genome. They are characterized by seven transmembrane domains (TMDs) with an intracellular N-terminus and an extracellular C-terminus (*Clyne et al., 1999*). During chemosensory signal transduction, ORs serve as signal-transducers by converting chemical signals into electric signals in the insect olfactory process (*Hallem & Carlson, 2006*). Functional insect ORs consist of a ligand-gated nonselective cation channel, composed of one divergent OR subunit that confers odor specificity, and one conserved OR co-receptor (ORco) subunit (*Sato et al., 2008*; *Butterwick et al., 2018*). The OR repertoire size is considerably variable among insect species, and the genes are rapidly evolving (*Andersson, Löfstedt & Newcomb, 2015*; *Robertson, 2019*).

IRs belong to the ionotropic glutamate receptor family (iGluRs). Their molecular structures consist of an extracellular ligand-binding domain with two lobes and an ion channel (*Benton et al., 2009*). IRs can be divided into two subfamilies: conserved 'antennal IRs', involved in olfaction, thermo- and hygrosensation, and species-specific 'divergent IRs', found in peripheral and internal gustatory organs and involved in gustation (*Croset et al., 2010*; *Koh et al., 2014*). Similar to ORco, IRs have highly conserved IR co-receptors in different insect species, including IR8a, IR25a and IR76b, which are widely expressed and play different roles in the process of odorant and taste sensation (*Abuin et al., 2011*). Besides the perception of acids, amines and amino acids (*Hussain et al., 2016*), IRs are also involved in sensing temperature and humidity stimuli, auditory function and regulating the circadian clock (*Senthilan et al., 2012*; *Chen et al., 2015*; *Ni et al., 2016*; *Frank et al., 2017*). To date, IRs function reported in lepidopteran species is quite limited. However, recent research showed that IR8a was essential for acid-mediated feces avoidance in ovipositioning hawkmoth, *Manduca sexta* (*Zhang et al., 2019*).

GRs and ORs have a similar membrane topology. GRs are mainly expressed in the gustatory organs (*Dunipace et al., 2001*; *Touhara & Vosshall, 2009*; *Guo et al., 2017*), although some genes are also expressed in insect antennae (*Croset et al., 2010*). Generally, GRs are not highly conserved among insect species, and mainly detect some nonvolatile

compounds and contact stimuli (*Clyne, Warr & Carlson, 2000*; *Scott et al., 2001*). However, candidate carbon dioxide ($CO_2$) receptors are highly conserved in lepidopteran species, such as BmorGR1-3 receptor from *Bombyx mori* (*Wanner & Robertson, 2008*), HmelGR1-3 from *Heliconius melpomene* (*Briscoe et al., 2013*) and HarmGR1-3 from *Helicoverpa armigera* (*Ning et al., 2016*).

SNMPs belong to the cluster of differentiation 36 (CD36) receptor family with two TMDs and are mainly expressed in pheromone sensitive neurons in *Drosophila* and moths (*Rogers, Krieger & Vogt, 2001*; *Benton, Vannice & Vosshall, 2007*; *Vogt et al., 2009*). SNMPs are divided into SNMP1 and SNMP2 subgroups (*Forstner et al., 2008*). Notably, SNMPs are known to participate in pheromone detection (*Jin, Ha & Smith, 2008*; *Pregitzer et al., 2014*; *Sun et al., 2019*), although the molecular mechanisms of their involvement in chemoreception are unknown.

Although various olfactory related genes have recently been identified in numerous moth species, most of them are from nocturnal moths, such as Noctuidae and Pyralidae (*Yuvaraj et al., 2018*). However, some moth species are diurnal, such as Castniidae, Phaudidae, Sphingidae and Zygaenidae (*Subchev, 2014*; *Monteys et al., 2016*; *Zheng et al., 2019*; *Chen et al., 2020*). Their activities are only observed during the daytime. Studies investigating the chemosensory genes in diurnal moths are scarce. These genes have only been studied in one non-ditrysian moth, *Eriocrania semipurpurella* (Eriocraniidae), belonging to the oldest lineages of Lepidoptera (*Yuvaraj et al., 2017*). Visual and/or olfactory cues mediate the orientation of most lepidopteran adults during mating. The contrasting lifestyles between the diurnal and nocturnal insects result in a substantial distinction in their sensory biology: diurnal butterflies primarily rely on visual cues while nocturnal moths largely rely on olfactory signals (*Martin et al., 2011*; *Arikawa, 2017*). However, this does not imply that butterflies are anosmic or moths are blind. Both butterflies and moths employ olfactory signals for sex communication. Besides, even at night, male moths strongly rely on visual cues from the landscape to track a pheromone trail to find a receptive female (*Preiss & Kramer, 1986*). While diurnal moths search for mating partners using olfactory cues over relatively long distances, they employ visual and auditory cues within short distances or even a combination of visual, olfactory and auditory signals (*Chen et al., 2020*).

Nevertheless, there has been a notable chemosensory/visual shift between moths and butterflies that at least affect long-range/short-range sex-attraction/courtship, as well as certain aspects of host plant recognition (*Costanzo & Monteiro, 2007*). A previous study of gene gain and loss within the general odorant binding proteins (GOBP)/pheromone binding proteins (PBP) of two moths and butterflies revealed that diurnal butterflies might have lost a PBP gene compared with nocturnal moths (*Vogt, Grosse-Wilde & Zhou, 2015*). Moreover, in our previous study, we constructed transcriptomes from both male and female adult antennae of *H. rhodope* using *de novo* transcriptome sequencing and assembly. We identified some chemosensory soluble proteins, such as odorant binding proteins (OBPs), chemosensory proteins (CSPs) and Niemann-Pick type C2 proteins (NPC2s) in this diurnal moth. We found that *H. rhodope* might also have lost a PBP gene (*Yang et al., 2020*). Therefore, we hypothesize that *H. rhodope* could also have lost

chemoreceptor genes associated with moth pheromone detection. However, the numbers, expression profiles and functions of chemoreceptor genes in *H. rhodope* are unknown.

In the present study, we identified candidate chemoreceptor genes based on our previous antennal transcriptomic data of *H. rhodope* (*Yang et al., 2020*). The phylogenetic relationships between the ORs, IRs, GRs and SNMPs of *H. rhodope* and other insect species were further investigated. Finally, we examined their tissue expression using quantitative real-time PCR (qRT-PCR). This work provides the basis for future functional characterization of olfactory genes in *H. rhodope.*

## MATERIALS & METHODS

### Insect rearing and tissue collection

*H. rhodope* was collected from *B. polycarpa* forest in Sui and Tang Dynasties City Ruins Botanical Garden (112.45°E, 34.64°N) in Luoyang city, Henan province, China. We received verbal permission to collect insects in Sui and Tang Dynasties City Ruins Botanical Garden from Yi Zhao, the deputy dean of the Garden. The larvae were reared on fresh *B. polycarpa* leaves under constant conditions of $25 \pm 1$ °C and $70 \pm 5\%$ relative humidity with 16 h light: 8 h dark photoperiod. Adult moths were grouped based on their sex and fed with 10% honey solution after emergence. For RNA-seq samples, 500 pairs of 1 day old adult antennae from each gender were dissected. For qRT-PCR analysis, we collected 50 male antennae (MA), 50 female antennae (FA), 50 heads (♀: ♂ = 1:1; H) whose antennae were cut off and 50 legs (♀: ♂ = 1:1; L) in three replications from 1 day old virgin adults. After collection, the samples were preserved in liquid nitrogen for further use.

### Identification of putative chemosensory receptor proteins

The antenna transcriptomes of unmated *H. rhodope* were reported in our previous study. Six female and male antennae samples (three biological replicates each gender) were separately sequenced using the Illumina Hiseq4000 platform. The female and male antennae yielded at least 21.62 and 21.39 million clean reads per sample, respectively. All clean reads from male and female samples were combined into an assembly that generated 50,218 unigenes with a mean length of 1,031.08 bp and an N50 length of 2,247 bp (*Yang et al., 2020*). A tBLASTn analysis was performed using available ORs, IRs, GRs and SNMPs protein sequences from lepidopteran species as 'queries' to identify candidate unigenes in *H. rhodope.* All putative ORs, IRs, GRs and SNMPs were manually checked using the BLASTx program in the National Center for Biotechnology Information (NCBI) with a cut-off $E$-value of $10^{-5}$. The OR genes were numbered arbitrarily, while the IR, GR and SNMP genes were named based on the highest scoring Blastx match from the NR database. Open reading frames (ORFs) were predicted using the ORF Finder (https://www.ncbi.nlm.nih.gov/orffinder/). The TMDs of the putative chemoreceptor genes were predicted on the TOPCONS website (http://topcons.net/).

### Comparison of transcript abundances of chemosensory receptor genes

Using the assembled transcriptome as reference sequences, the clean data from various samples of *H. rhodope* were mapped back onto the reference sequences using Bowtie2 v2.1.0

software (*Langmead et al., 2009*). The unigene expression levels among various samples were estimated with RSEM (*Li & Dewey, 2011*) according to the read count values of the unigenes for each sample, which were obtained from the mapping results. The transcript abundance of candidate chemoreceptor genes in male and female antennal transcriptomes was calculated using fragments per kb per million fragments (FPKM) values (*Li & Dewey, 2011*). Heatmap plots of the chemoreceptor gene expression were generated in Microsoft Excel, using the conditional formatting option. For each plot, blue color was used for the minimum value, yellow for the midpoint and red for the maximum value. For all gene families, the range was specified for each tissue type independently, such that the color gradient was set based upon the highest FPKM values within each tissue, not across all tissues. The criteria for estimating significant differentially expressed genes were set as the absolute value of log2 Fold Change >1.

## Phylogenetic analysis

Phylogenetic trees of candidate chemoreceptors were constructed using MEGA7.0 software (*Kumar, Stecher & Tamura, 2016*). Amino acid sequences were first aligned using the program ClustalX (http://www.clustal.org/clustal2/). Maximum likelihood (ML) statistical method was used to infer evolutionary relationship and unrooted trees were built with the Jones-Taylor-Thornton (JTT) model used to obtain the initial trees for the heuristic search. The tree calculation was carried out using 1,000 bootstrap replicates. Generated trees were edited using Figtree software (http://tree.bio.ed.ac.uk/software/figtree/). The protein sequences of chemoreceptors used for building phylogenetic trees are listed in File S1.

## Expression level analysis

We selected 10 ORs, all IRs, 3 GRs and 2 SNMPs to verify their expression profiles. 10 ORs and 3 GRs were selected for qRT-PCR analysis because they were significant DEGs between male and female antennae according to the criteria as described above. The relative expression levels of these genes in different tissues were determined using qRT-PCR. Different tissues were collected from both male and female adults as described above. Total RNA was extracted using TRIzol reagent. First-strand cDNA was synthesized from 2 μg of total RNA using the PrimeScript RT reagent kit with gDNA Eraser (Takara, Dalian, China) according to the manufacturer's instructions. qRT-PCR primers (Table S1) were designed with Primer Premier 5 (Premier Biosoft International, CA, USA) and synthesized by Sangon Biotech Co., Ltd (Shanghai, China). The glyceraldehyde-phosphate dehydrogenase (GAPDH) gene was identified from the *H. rhodope* antennal transcriptome and used as the internal reference (*Yang et al., 2020*). The qRT-PCR was conducted on a StepOne Plus Real-time PCR System (Applied Biosystems, Foster City, CA, USA) using SYBR Premix ExTaq II (Tli RNaseH Plus) (Takara, Dalian, China). Each reaction (20 μL volume) contained 2 μL cDNA, 10 μL SYBR® Premix Ex Taq, 0.4 μL forward and reverse primers (10 μM), and 7.2 μL RNase-free double distilled water. qRT-PCR was performed as follows: initial denaturation at 95 °C for 3 min, 40 cycles at 95 °C for 10 s and 60 °C for 30 s. Melting curve analysis was performed from 55 °C to 95 °C to determine the

specificity of qPCR primers. To determine the efficiency of the qPCR primers, a standard curve (cDNA concentration vs. Ct) was produced with a 5-fold dilution series of legs cDNA corresponding to one microgram total RNA. qRT-PCR efficiencies were then calculated according to the equation: $E = (10^{-[1/slope]} - 1)*100$ (*Pfaffl, 2001*; *Radonić et al., 2004*). The $2^{-\Delta\Delta Ct}$ method was used to analyze gene expression profiles (*Pfaffl, 2001*). All data were normalized to endogenous GAPDH rRNA levels from the identical tissue sample, and the relative fold change in the different tissues was calculated with the transcript level of the legs as the calibrator. Each reaction was performed in triplicate (from three biological replicates).

Relative expressions of each test gene from the different tissues were compared using a one-way analysis of variance (ANOVA) in SPSS 20.0 software (IBM, Chicago, IL, USA) followed by the least significant difference test (LSD) (critical values corresponding to $P = 0.05$).

## RESULTS

### Identification and phylogenetic analyses of ORs in *H. rhodope*

A total of 45 candidate ORs, including 44 conventional ORs and one ORco, were identified in *H. rhodope* antennal transcriptome (Table 1). Among these genes, 25 candidate OR genes had 7 TMDs and full ORFs encoding more than 370 amino acids. HrhoORco shared 85.44% sequence identity with *Galleria mellonella* ORco (NCBI ID: QEI46859) and 84.81% sequence identity with *Ostrinia furnacalis* ORco (NCBI ID: XP_028178675). Gene expression levels of all 45 ORs were assessed using the FPKM values, where 23 *ORs* were relatively highly expressed in the FA and two *ORs* were relatively highly expressed in the MA. The remaining 20 ORs showed no differences in expression levels between sexes (Fig. 1A).

A phylogenetic tree was constructed with 195 ORs from *H. rhodope*, *B. mori*, *H. armigera*, *O. furnacalis*, *Cnaphalocrocis medinalis*, *Heliothis virescens*, *Dendrolimus punctatus*, *Spodoptera littoralis* and *M. sexta*. Most of the HrhoORs have orthologous relationships with the other lepidopteran species (Fig. 2). As anticipated, HrhoORco clustered in the ORco family. In addition, four HrhoORs (HrhoOR4, 16, 39 and 40) were segregated into one unique clade whereas HrhoOR15/23, HrhoOR2/32 and HrhoOR21/28 clustered together. Interestingly, *H. rhodope* ORs lacked in the classical moth pheromone receptor (PR) clade, while HrhhoOR14 and HrhoOR30 clustered within a novel PR clade containing SlitOR5 in *S. littoralis* and several candidate PRs from *D. punctatus*. These ORs were recently identified as a novel lineage of PRs detecting type I pheromones in Lepidoptera (*Bastin-Héline et al., 2019*; *Shen et al., 2020*).

### Identification and phylogenetic analyses of IRs in *H. rhodope*

Nine putative IRs were identified from the antennal transcriptome of *H. rhodope* and named as HrhoIR8a, 21a, 76b, 41a, 60a, 75q2, 68a, 75p and 40a based on homologous sequences from other insects (Table 2). All candidate IRs were partial sequences and the TMDs of IRs ranged from 1 to 4.Three IRs (*HrhoIR8a*, *60a* and *75q.2*) were significantly

Yang et al. (2020), *PeerJ*, DOI 10.7717/peerj.10035

**Table 1  The best Blast match of candidate odorant receptors (ORs) in *H. rhodope*.**

| Accession number | Gene name | ORF (bp) | Complete ORF | TMD (No.) | Blastx annotation (Description/Species) | Accession number | E-value | Identity (%) |
|---|---|---|---|---|---|---|---|---|
| MN515166 | HrhoOrco | 1422 | Y | 7 | Orco [*Galleria mellonella*] | QEI46859.1 | 0 | 85.44 |
| MN515167 | HrhoOR1 | 687 | N | 4 | Olfactory receptor 11 [*Ctenopseustis obliquana*] | AIT71985.1 | 5.00E−40 | 36.65 |
| MN515168 | HrhoOR2 | 978 | N | 6 | Putative odorant receptor OR57 [*Cydia nigricana*] | AST36406.1 | 2.00E−57 | 34.97 |
| MN515169 | HrhoOR3 | 1215 | Y | 7 | Odorant receptor [*Eogystia hippophaecolus*] | AOG12952.1 | 1.00E−149 | 51.36 |
| MN515170 | HrhoOR4 | 1203 | Y | 7 | Olfactory receptor 66 [*Ctenopseustis herana*] | AIT69908.1 | 3.00E−79 | 35.09 |
| MN515171 | HrhoOR5a | 669 | N | 4 | Odorant receptor 94a-like isoform X2 [*Manduca sexta*] | XP_030035689.1 | 8.00E−47 | 47.32 |
| MN515172 | HrhoOR5b | 603 | N | 2 | Putative olfactory receptor 21 [*Ostrinia furnacalis*] | BAR43463.1 | 2.00E−86 | 64.14 |
| MN515173 | HrhoOR6 | 1212 | Y | 7 | Putative odorant receptor OR27 [*Hedya nubiferana*] | AST36262.1 | 0 | 67.33 |
| MN515174 | HrhoOR7 | 1224 | Y | 7 | Odorant receptor [*Eogystia hippophaecolus*] | AOG12916.1 | 1.00E−135 | 49.63 |
| MN515175 | HrhoOR8 | 714 | N | 3 | Odorant receptor [*Eogystia hippophaecolus*] | AOG12915.1 | 3.00E−102 | 63.14 |
| MN515176 | HrhoOR9 | 417 | N | 3 | Putative odorant receptor 92a [*Bombyx mandarina*] | XP_028031179.1 | 1.00E−22 | 36.96 |
| MN515177 | HrhoOR10 | 1341 | Y | 7 | Odorant receptor 1 [*Cnaphalocrocis medinalis*] | ALT31655.1 | 0 | 63.33 |
| MN515178 | HrhoOR11 | 1185 | Y | 7 | Odorant receptor 85c-like [*Vanessa tameamea*] | XP_026493935.1 | 3.00E−88 | 38.66 |
| MN515179 | HrhoOR12 | 792 | N | 6 | Odorant receptor [*Eogystia hippophaecolus*] | AOG12948.1 | 1.00E−143 | 74.62 |
| MN515180 | HrhoOR13 | 1152 | Y | 7 | Odorant receptors OR25 [*Lobesia botrana*] | AXF48775.1 | 1.00E−158 | 56.49 |
| MN515181 | HrhoOR14 | 1206 | Y | 7 | Putative odorant receptor OR30 [*Cydia pomonella*] | AFC91738.2 | 7.00E−122 | 44.56 |
| MN515182 | HrhoOR15 | 378 | N | 3 | Putative odorant receptor 57 [*Conopomorpha sinensis*] | AXY83406.1 | 1.00E−30 | 47.54 |
| MN515183 | HrhoOR16 | 1188 | Y | 7 | Olfactory receptor 66 [*Ctenopseustis herana*] | AIT69908.1 | 2.00E−97 | 40.54 |
| MN515184 | HrhoOR17 | 1263 | Y | 7 | Putative odorant receptor OR13 [*Hedya nubiferana*] | AST36253.1 | 9.00E−156 | 55.81 |
| MN515185 | HrhoOR18 | 1176 | Y | 7 | Odorant receptor [*Eogystia hippophaecolus*] | AOG12907.1 | 0 | 67.70 |
| MN515186 | HrhoOR19 | 1302 | Y | 7 | putative odorant receptor OR63 [*Hedya nubiferana*] | AST36285.1 | 1.00E−154 | 49.07 |
| MN515187 | HrhoOR20 | 951 | N | 4 | odorant receptor [*Eogystia hippophaecolus*] | AOG12933.1 | 2.00E−119 | 54.78 |
| MN515188 | HrhoOR21 | 1233 | Y | 7 | olfactory receptor 11 [*Ctenopseustis herana*] | AIT69876.1 | 6.00E−101 | 40.35 |
| MN515189 | HrhoOR22 | 1200 | Y | 7 | olfactory receptor 20 [*Helicoverpa armigera*] | ACC63240.1 | 6.00E−121 | 44.99 |
| MN515190 | HrhoOR23 | 1110 | Y | 7 | olfactory receptor OR54 [*Planotortrix octo*] | AJF23812.1 | 2.00E−145 | 54.30 |

**Table 1** (*continued*)

| Accession number | Gene name | ORF (bp) | Complete ORF | TMD (No.) | Blastx annotation (Description/Species) | Accession number | E-value | Identity (%) |
|---|---|---|---|---|---|---|---|---|
| MN515191 | HrhoOR24 | 1212 | Y | 7 | olfactory receptor 27 [*Helicoverpa armigera*] | ARF06962.1 | 8.00E−115 | 45.18 |
| MN515192 | HrhoOR25 | 1197 | Y | 7 | odorant receptor [*Eogystia hippophaecolus*] | AOG12927.1 | 0 | 71.43 |
| MN515193 | HrhoOR26 | 1332 | Y | 7 | olfactory receptor 71 [*Ctenopseustis herana*] | AIT69911.1 | 0 | 65.08 |
| MN515194 | HrhoOR27 | 378 | N | 2 | olfactory receptor 37 [*Carposina sasakii*] | AYD42255.1 | 2.00E−42 | 51.22 |
| MN515195 | HrhoOR28 | 352 | N | 2 | putative olfactory receptor 25 [*Ostrinia furnacalis*] | BAR43467.1 | 4.00E−22 | 36.75 |
| MN515196 | HrhoOR29 | 1230 | Y | 7 | odorant receptor 24a-like [*Ostrinia furnacalis*] | XP_028158823.1 | 0 | 71.15 |
| MN515197 | HrhoOR30 | 1194 | N | 6 | odorant receptor [*Eogystia hippophaecolus*] | AOG12934.1 | 3.00E−91 | 36.57 |
| MN515198 | HrhoOR31 | 1284 | Y | 7 | odorant receptor [*Eogystia hippophaecolus*] | AOG12926.1 | 0 | 75.93 |
| MN515199 | HrhoOR32 | 927 | N | 3 | putative odorant receptor OR57 [*Cydia nigricana*] | AST36406.1 | 3.00E−47 | 33.00 |
| MN515200 | HrhoOR33 | 756 | N | 5 | olfactory receptor 14 [*Heortia vitessoides*] | AZB49428.1 | 7.00E−84 | 53.57 |
| MN515201 | HrhoOR34 | 1212 | Y | 7 | odorant receptors OR29 [*Lobesia botrana*] | AXF48779.1 | 0 | 65.58 |
| MN515202 | HrhoOR35 | 612 | N | 3 | Olfactory receptor 23 [*Manduca sexta*] | CUQ99405.1 | 5.00E−64 | 52.24 |
| MN515203 | HrhoOR36 | 1218 | Y | 7 | odorant receptor [*Eogystia hippophaecolus*] | AOG12941.1 | 2.00E−134 | 46.56 |
| MN515204 | HrhoOR37 | 1143 | Y | 7 | olfactory receptor 32 [*Cnaphalocrocis medinalis*] | ANZ03145.1 | 4.00E−159 | 56.74 |
| MN515205 | HrhoOR38 | 1215 | N | 0 | putative odorant receptor [*Peridroma saucia*] | AVF19641.1 | 0 | 67.48 |
| MN515206 | HrhoOR39 | 1005 | N | 6 | Olfactory receptor 65 [*Manduca sexta*] | CUQ99419.1 | 1.00E−69 | 38.74 |
| MN515207 | HrhoOR40 | 489 | N | 3 | putative odorant receptor OR23 [*Athetis lepigone*] | AOE48028.1 | 3.00E−41 | 43.90 |
| MN515208 | HrhoOR41 | 1335 | Y | 7 | gustatory receptor 6 [*Helicoverpa armigera*] | ASW18695.1 | 0 | 67.59 |
| MN515209 | HrhoOR42 | 720 | N | 1 | putative odorant response protein ODR-4 [*Danaus plexippus plexippus*] | OWR50364.1 | 2.00E−144 | 84.45 |
| MN515210 | HrhoOR43 | 526 | N | 3 | Odorant receptor [*Eogystia hippophaecolus*] | AOG12949.1 | 2.00E−28 | 37.42 |

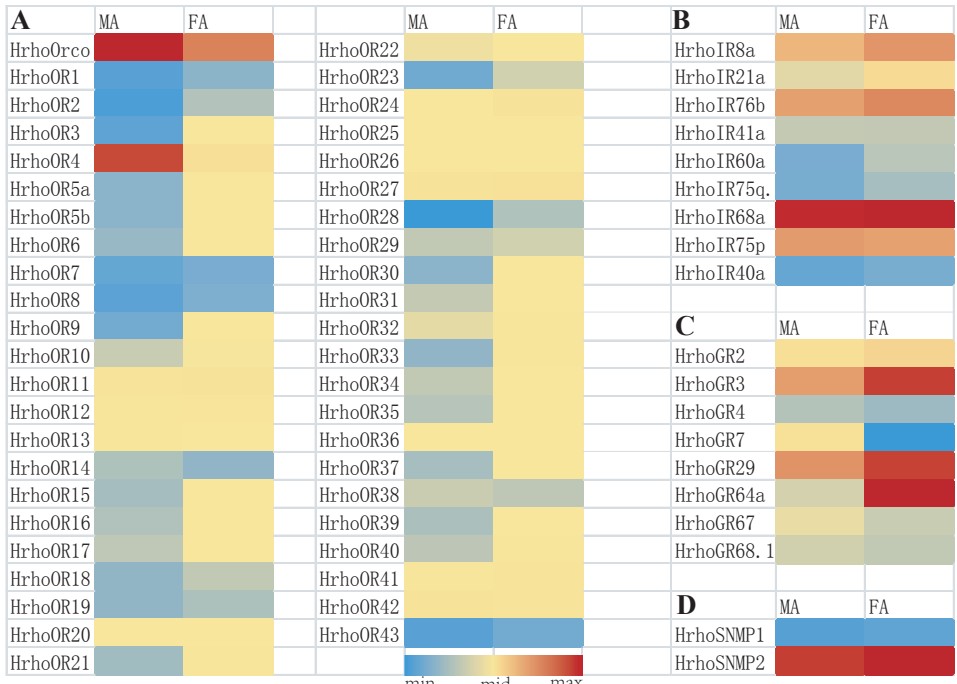

**Figure 1** **Heat map of FPKM values for ORs, IRs, GRs and SNMPs in male antennae (MA) and female antennae (FA).** Blue color indicates low expression, yellow color indicates moderate expression and red color indicates high expression. FA, female antennae; MA, male antennae. (A) ORs, odorant receptors, (B) IRs, ionotropic receptors, (C) GRs, gustatory receptors, (D) SNMPs, sensory neuron membrane proteins.

higher expressed in FA than in MA. The highest FPKM value of IRs (*HrhoIR68a*, FPKM = 202.80) was found in the FA (Fig. 1B).

A phylogenetic tree was constructed with 99 IRs from *H. rhodope*, *Cydia pomonella*, *Epiphyas postvittana* and *D. melanogaster* (Fig. 3). We found that HhroIR8a, HrhoIR76b, HrhoIR75p, HrhoIR75q.2, HrhoIR21a, HrhoIR68a, HrhoIR40a, HhroIR60 and HrhoIR41a were clustered into IR8a, IR76b, IR75, IR21a, IR68a, IR40a, IR60a and IR41a clades with high bootstrap values, respectively. According to their positions in the phylogenetic tree and based on the strong bootstrap support, the candidate HrhoIRs names were consistent with the number and suffix of known IRs. However, no orthologs for IR25a, IR75d, IR93a and IR87a were identified from *H. rhodope*.

## Identification and phylogenetic analyses of GRs in *H. rhodope*

We identified eight candidate GR genes in the antennal transcriptomes of *H. rhodope* (Table 3). All GR genes had incomplete ORFs ranging from 390 to 1218 bp in length with 1–5 TMDs. The FPKM values of all *HrhoGRs* were less than those of OR, IR and SNMP genes. Three GR genes (*HrhoGR3*, *HrhoIR29* and *HrhoIR64a*) were significantly higher expressed in the FA than in the MA (Fig. 1C).

Phylogenetic analysis of 81 GRs from five lepidopteran species showed that several candidate GRs were closely related to other known insect GRs that function as sugar

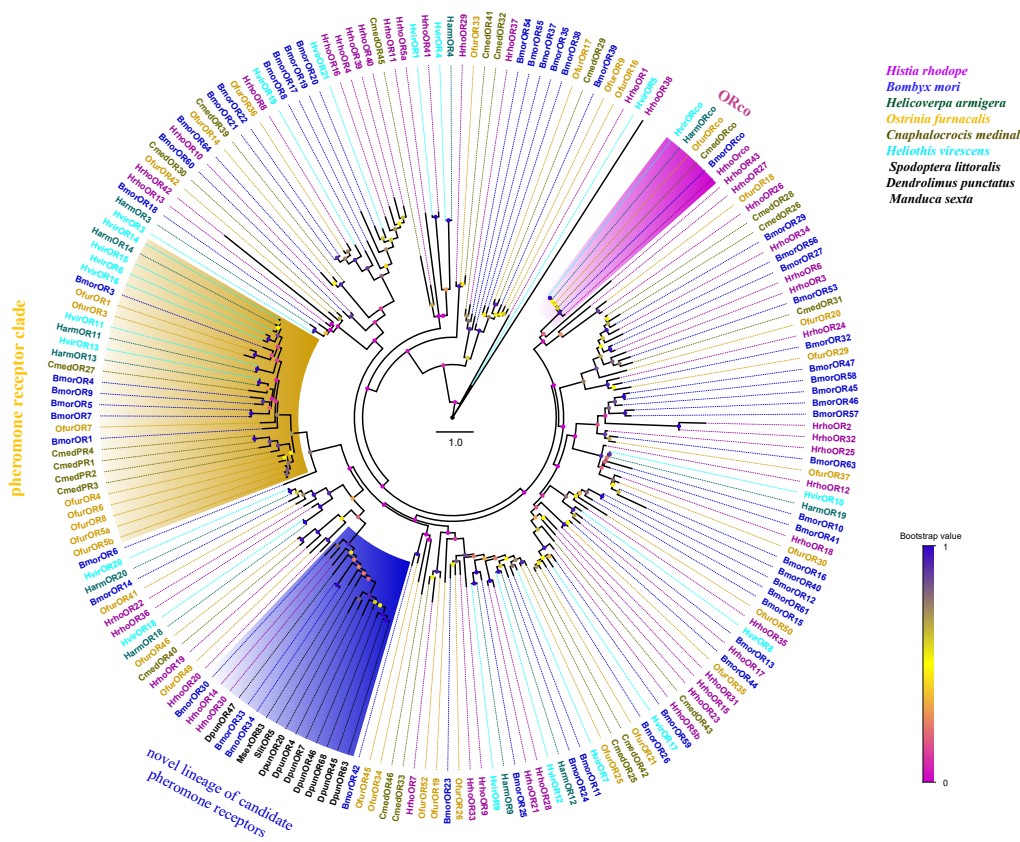

**Figure 2 Phylogenetic tree of putative odorant receptors (ORs).** Branch support (circles at the branch nodes) was estimated using an approximate likelihood ratio test based on the scale indicated at the bottom right. The clade of traditional PR clade is marked in yellow, the novel lineage of candidate PR clade is marked in blue and ORco in purple.

receptors, bitter receptors and $CO_2$ receptors. HrhoGR7 and HhroGR64a clustered into sugar receptor clade, HrhoGR67 clustered into bitter receptor subfamily and HrhoGR2 clustered into the $CO_2$ receptor clade. Notably, the other four GRs (HrhoGR3/4/29/68.1) had orthologous relationships with *C. pomonella* (Fig. 4).

### Identification and phylogenetic analyses of SNMPs in *H. rhodope*

We identified two candidate SNMPs (HrhoSNMP1 and HrhoSNMP2) from antennal transcriptomes of *H. rhodope* (Table 4). Two SNMP genes had intact ORFs (length: 1,422-1,581 bp) with 1–2 TMDs. According to the FPKM values of SNMPs, *HrhoSNMP1* and *HrhoSNMP2* were equally expressed in both the FA and the MA groups (Fig. 1D). SNMPs phylogenetic tree revealed that HrhoSNMP1 and HrhoSNMP2 clustered into SNMP1 and SNMP2 clades, respectively (Fig. 5).

### Tissue-specific expression analysis by qRT-PCR

To assess the difference in expression of chemoreceptor genes between male and female antennae, heads (without antennae) and legs and test the RNA-Seq results, 10 ORs, all IRs, 3 GRs and 2 SNMPs were selected for qRT-PCR. The results showed that the expression levels

**Table 2** **The best Blast match of candidate ionotrpic receptors (IRs) in *H. rhodope*.**

| Accession number | Gene name | ORF (bp) | Complete ORF | TMD (No.) | Blastx annotation (Description/Species) | Accession number | E-value | Identity (%) |
|---|---|---|---|---|---|---|---|---|
| MN515211 | HrhoIR8a | 2631 | N | 4 | Ionotropic receptor 8a [*Ostrinia furnacalis*] | BAR64796.1 | 0 | 73.84 |
| MN515212 | HrhoIR21a | 1182 | N | 2 | Ionotropic receptor 21a [*Eogystia hippophaecolus*] | AOG12851.1 | 0 | 68.77 |
| MN515213 | HrhoIR76b | 693 | N | 1 | Putative ionotropic receptor IR76b [*Cydia nigricana*] | AQM73618.1 | 4E−99 | 63.22 |
| MN515214 | HrhoIR41a | 1128 | N | 1 | Ionotropic receptor 41a [*Ostrinia furnacalis*] | BAR64800.1 | 2E−142 | 56.75 |
| MN515215 | HrhoIR60a | 1959 | N | 3 | Ionotropic receptor IR60a [*Cnaphalocrocis medinalis*] | APY22698.1 | 0 | 49.92 |
| MN515216 | HrhoIR75q.2 | 1686 | N | 3 | Putative ionotropic receptor IR75q.2 [*Cydia fagiglandana*] | AST36364.1 | 0 | 62.14 |
| MN515217 | HrhoIR68a | 1302 | N | 4 | Ionotropic receptor 68a [*Eogystia hippophaecolus*] | AOG12853.1 | 0 | 74.88 |
| MN515218 | HrhoIR75p | 1707 | N | 2 | Ionotropic receptor 75a-like [*Bombyx mandarina*] | XP_028025288.1 | 0 | 63.46 |
| MN515219 | HrhoIR40a | 1632 | N | 2 | Ionotropic receptor 40a [*Manduca sexta*] | XP_030034643.1 | 0 | 79.96 |

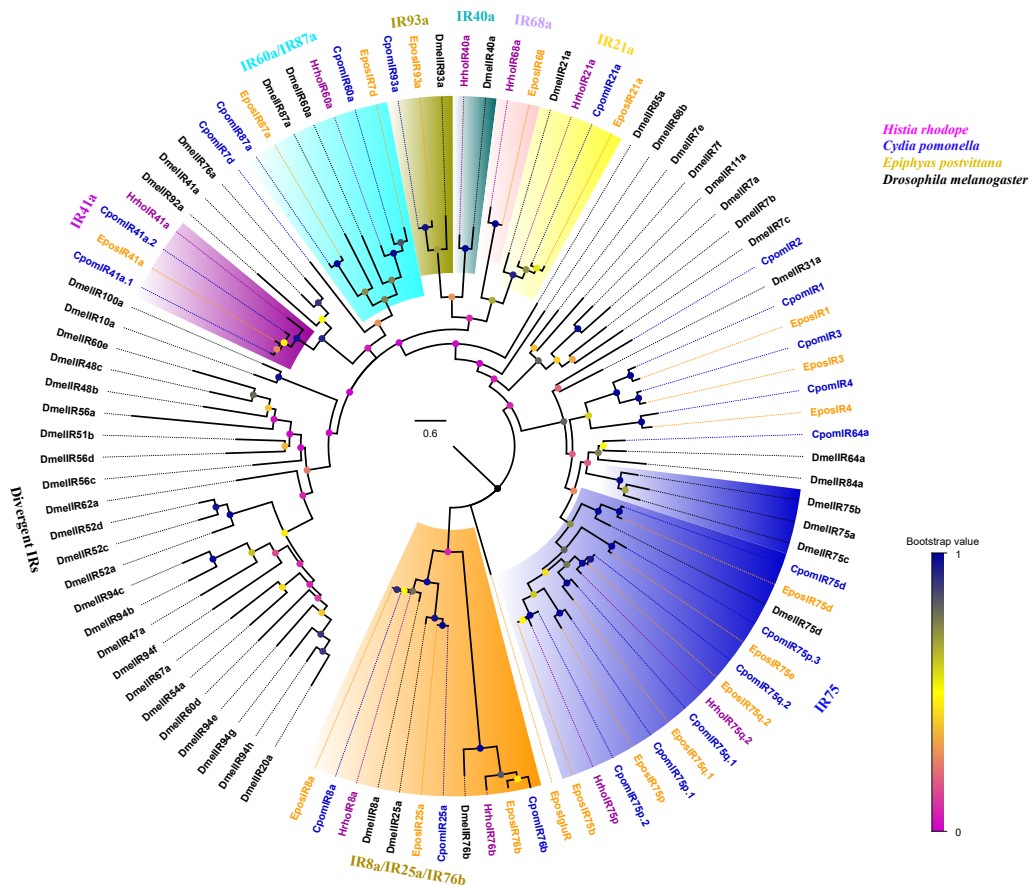

**Figure 3** **Phylogenetic tree of putative ionotropic receptors (IRs).** Branch support (circles at the branch nodes) was estimated using an approximate likelihood ratio test based on the scale indicated at the bottom right. Branches containing putative IR co-receptors and conversed antennal IRs are colored.

of the tested genes in the male and female antennae were consistent with the RNA-Seq results. The qRT-PCR results revealed that all 10 *HrhoORs* were significantly higher expressed in the antennae than in the heads and legs. Furthermore, 6 OR genes (*HrhoOR9, 24, 26, 33, 37* and *40*) were significantly higher expressed in the FA while *HrhoOR4* and *HrhoOR13* were significantly higher expressed in the MA. Interestingly, *HrhoORco* and *HrhoOR11* showed no differences in expression between sexes (Figs. 6A–6J). Notably, *HrhoORco, HrhoOR4* and *HrhoOR37* showed higher expression levels in the antennae compared with the expression levels of other *HrhoORs*.

The qRT-PCR results revealed that the expression of 5 IRs (*HrhoIR8a, 21a, 40a, 60a* and *76b*) was significantly higher in the FA while that of 3 genes (*HrhoIR41a, 68a* and *75p*) was equally expressed in the antennae of both sexes. Additionally, we found that *HrhoIR75q.2* and *HrhoIR40* had higher expression in the heads compared with other genes (Figs. 6K–6S).

*HrhoGR2* and *HrhoGR64a* displayed significant FA-specific expressions, while *HrhoGR67* had significantly higher expression in the heads than in other tissues (Figs. 6T–6V).

Peerj

**Table 3** **The best BLAST match of candidate gustatory receptors (GRs) in *H. rhodope*.**

| Accession number | Gene name | ORF (bp) | Complete ORF | TMD (No.) | Blastx annotation (Description/Species) | Accession number | E-value | Identity (%) |
|---|---|---|---|---|---|---|---|---|
| MN515222 | HrhoGR2 | 612 | N | 4 | Putative gustatory receptor GR2 [*Hedya nubiferana*] | AST36211.1 | 7E−116 | 88.24 |
| MN515223 | HrhoGR3 | 390 | N | 2 | Antennal gustatory receptor 9 [*Dendrolimus punctatus*] | ARO70281.1 | 5E−36 | 50.38 |
| MN515224 | HrhoGR4 | 198 | N | 1 | Antennal gustatory receptor 9 [*Dendrolimus punctatus*] | ARO70281.1 | 7E−10 | 55.32 |
| Mn515229 | HrhoGR7 | 237 | N | 1 | Gustatory receptor 7 [*Operophtera brumata*] | KOB71247.1 | 1E−11 | 47.06 |
| MN515225 | HrhoGR29 | 393 | N | 2 | Putative gustatory receptor GR29 [*Cydia fagiglandana*] | AST36346.1 | 1E−29 | 45.45 |
| MN515226 | HrhoGR64a | 1218 | N | 5 | Gustatory receptor for sugar taste 64a-like [*Bombyx mandarina*] | XP_028037655.1 | 7E−153 | 53.50 |
| MN515227 | HrhoGR67 | 693 | N | 4 | Gustatory receptor 67 [*Bombyx mori*] | NP_001233216.1 | 4E−48 | 38.29 |
| MN515228 | HrhoGR68.1 | 879 | N | 4 | Putative gustatory receptor GR68.1 [*Hedya nubiferana*] | AST36219.1 | 1E−43 | 34.56 |

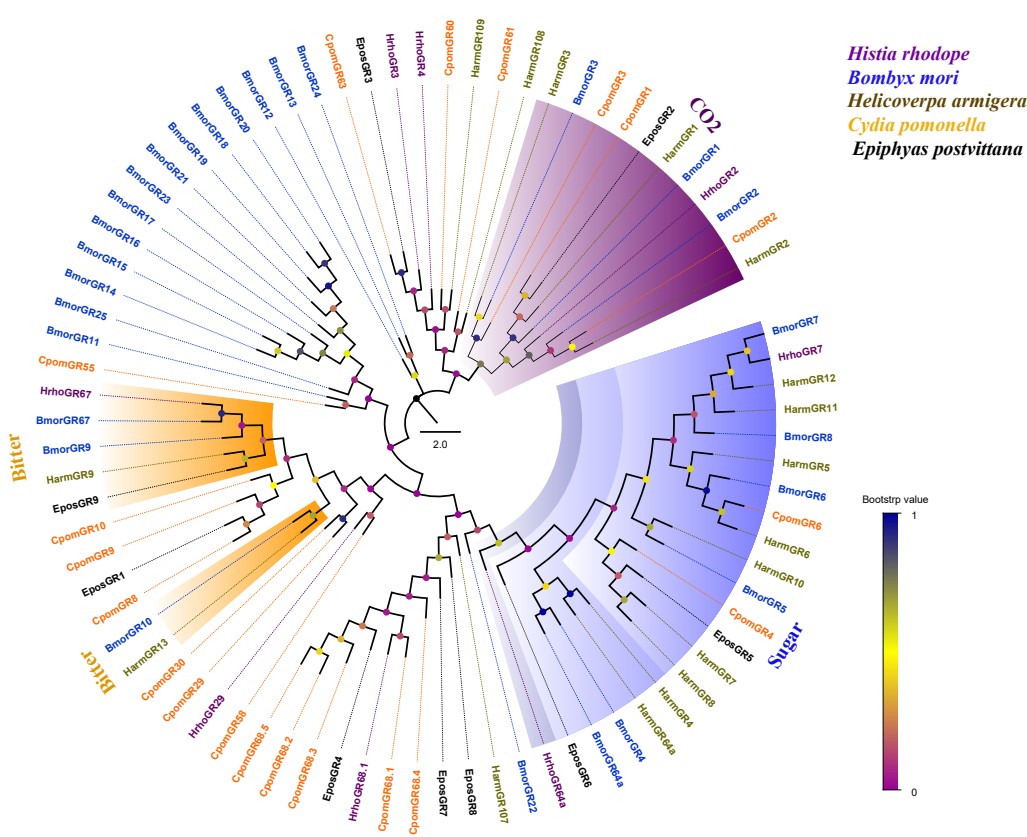

**Figure 4** **Phylogenetic tree of putative gustatory receptors (GRs).** Branch support (circles at the branch nodes) was estimated using an approximate likelihood ratio test based on the scale indicated at the bottom right. The sugar receptor clade, bitter receptor clade and CO2 receptor clade are colored.

*HrhoSNMP1* and *HrhoSNMP2* were significantly expressed at high levels in the antennae in both sexes; *HrhoSNMP2* was also highly expressed in heads (Figs. 6W–6X).

## DISCUSSION

To date, a growing number of chemosensory genes have been identified in moths. However, most of the genes are from nocturnal moths (Ditrysia) (*Yuvaraj et al., 2018*). Currently, the molecular basis of chemoreception in the Zygaenidae family of diurnal moths is poorly understood. The identification and characterization of the chemoreceptor genes of *H. rhodope*, an important forest pest and diurnal moth, will improve our understanding of olfaction mechanisms in Zygaenidae and provide the basis for further exploration of sensory disparities between the diurnal and nocturnal moths.

In the present study, we identified 45 ORs, 9 IRs, 8 GRs and 2 SNMPs from our previous *H. rhodope* transcriptomic data (*Yang et al., 2020*). The number of chemoreceptor genes in *H. rhodope* was less compared to previously reported numbers in other nocturnal lepidopteran species, such as 62 ORs, 20 IRs and 16 GRs in *Mythimna separata* (*Du et al., 2018*), 64 ORs, 22 IRs and 30 GRs in *Spodoptera exigua* (*Zhang et al., 2018*), 60 ORs, 21

Yang et al. (2020), *PeerJ*, DOI 10.7717/peerj.10035

**Table 4  The best BLAST match of candidate sensory neuron membrane proteins (SNMPs) in *H. rhodope*.**

| Accession number | Gene name | ORF (bp) | Complete ORF | TMD (No.) | Blastx annotation (Description/Species) | Accession number | E-value | Identity (%) |
|---|---|---|---|---|---|---|---|---|
| MN515220 | HrhoSNMP1 | 1581 | Y | 2 | Sensory neuron membrane protein 1 [*Papilio xuthus*] | KPI90909.1 | 0 | 66.22 |
| MN515221 | HrhoSNMP2 | 1422 | Y | 1 | Sensory neuron membrane protein 2 [*Ostrinia nubilalis*] | E5EZW9.1 | 0 | 64.08 |

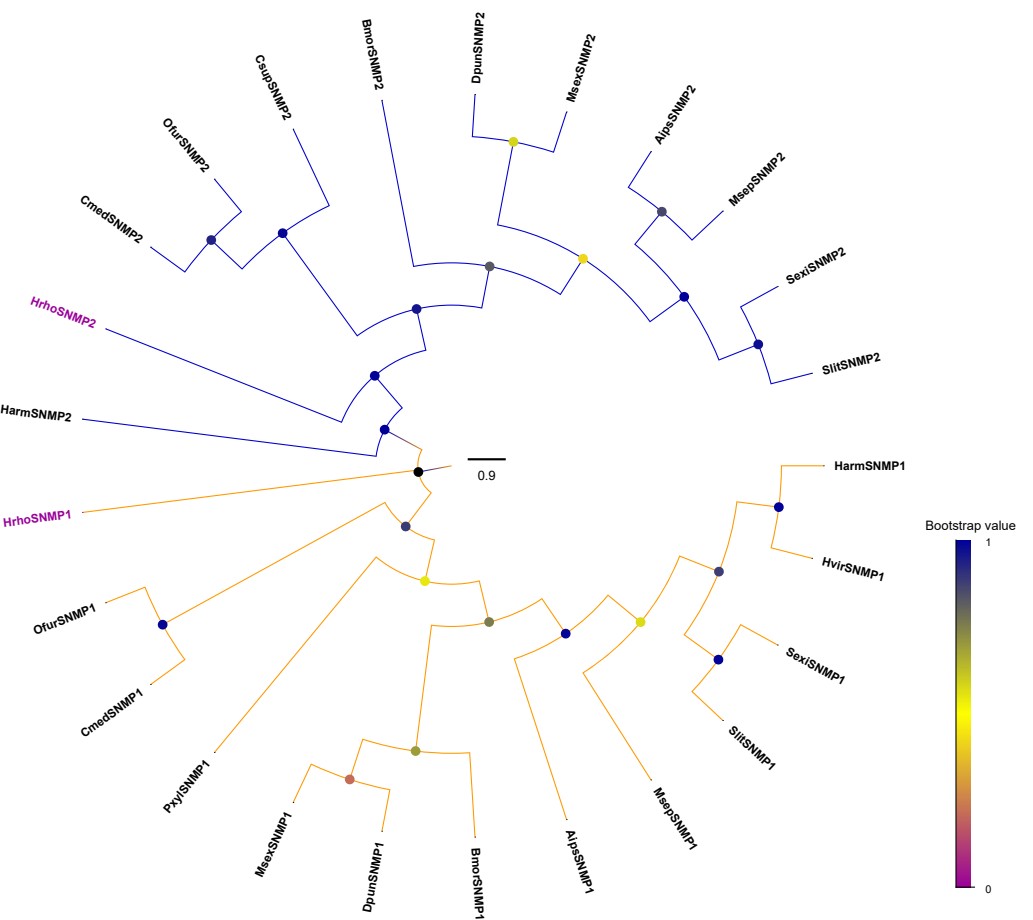

**Figure 5** **Phylogenetic tree of putative sensory neuron membrane proteins (SNMPs).** Branch support (circles at the branch nodes) was estimated using an approximate likelihood ratio test based on the scale indicated at the bottom right. The SNMP1 clade and SNMP2 clade are colored.

IRs and 197 GRs in *H. armigera* (*Liu et al., 2014*; *Xu et al., 2016*), 60 ORs, 17 IRs and 17 GRs in *S. littoralis* (*Walker III et al., 2019*) and 58 ORs, 21 IRs and 22 GRs in *C. pomonella* (*Walker III et al., 2016*). Previous studies reported that different chemosensory behavior (*Zhang et al., 2017*), life stage (*He et al., 2017*), ecological niche breadth (*Gouin et al., 2017*), phenotype (*Purandare & Brisson, 2020*), mating status (*Jin et al., 2017*), circadian rhythm (*Gadenne, Barrozo & Anton, 2016*) and feeding trait (*Taparia, Ignell & Hill, 2017*) influence the variation in chemosensory gene number. The smaller number of genes identified in the current study may be due to the following reasons. Firstly, we identified olfactory related genes from antenna transcriptome data only, whereas previous studies used transcriptome data from different developmental stages and organs, especially for GRs. Secondly, the number of chemosensory genes may be related to its ability for diverse host-odor detection to feed on the range size of host plants (*Gouin et al., 2017*). *H. rhodope* is an oligophagous pest requiring no additional olfactory proteins to perceive single host chemicals (*Gouin et al., 2017*; *Yang et al., 2020*). Finally, *H. rhodope* is a diurnal insect, although it is a moth

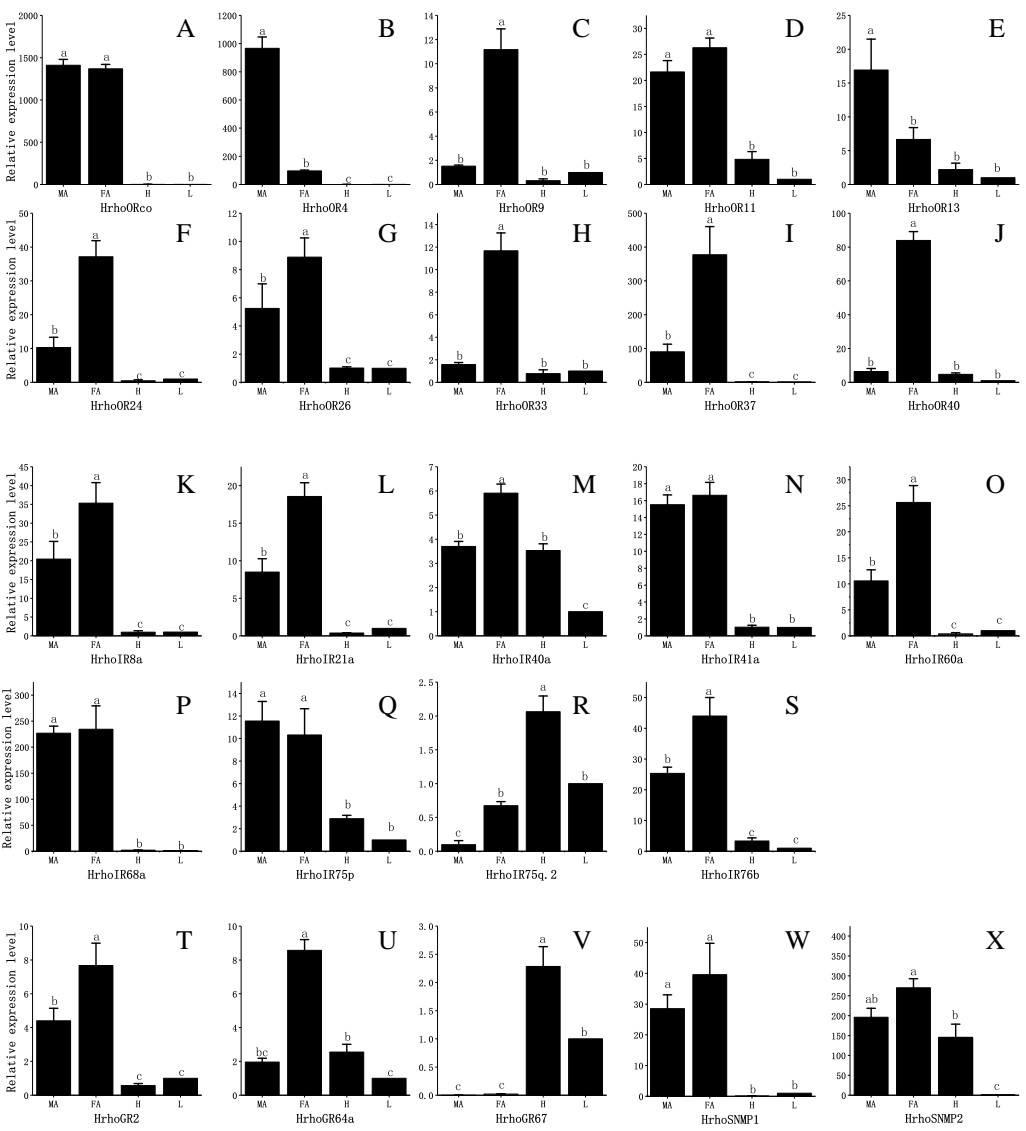

**Figure 6** **Relative expression levels of the chemoreceptor genes in the male antennae, female antennae, heads and legs from *H. rhodope*.** (A–J) ORs. (K–S) IRs. (T–V) GRs. (W–X) SNMPs. MA, male antennae; FA, female antennae; H, heads (without antennae); L, legs. The GAPDH was used as the reference gene, Gene expressions in various tissues are normalized relative to that in legs. Different lowercase letters mean significant difference between tissues ($p < 0.05$, ANOVA, LSD).

and not a butterfly. Diurnal moths mainly depend on visual cues while nocturnal moths mainly depend on their odorant signals (*Vogt, Grosse-Wilde & Zhou, 2015*). Chemosensory gene loss in diurnal moths compared to nocturnal moths could result from the shift from olfactory to visual communication.

Odorant receptors act as signal-transducers that convert chemical molecules to electrical signals in insects (*Leal, 2013*). ORs are desirable targets for alternative strategies to control insect populations due to their critical roles in the initial steps of the olfactory response

process (*Venthur & Zhou, 2018*). In this study, 44 specific ORs and 1 ORco were identified. ORco genes are widely expressed and highly conserved among lepidopteran species and play a key role in insect olfaction (*Touhara & Vosshall, 2009*). HrhoORco shared a high degree of similarity with the ORco genes of other insects and clustered into the ORco subfamily. Previous studies showed that many ORco mutants were anosmic. For instance, mutagenesis of BmolORco and MsexORco severely disrupted the olfactory system (*Liu et al., 2017a*; *Fandino et al., 2019*), suggesting that the ORco genes might be the primary potential target genes for pest management.

Generally, phylogenetic analysis of lepidopteran ORs showed two specific branches comprising the ORco orthologs and the traditional PR clade. Interestingly, no PRs were related to the conserved lepidopteran PR clade. However, two ORs (HrhoOR14 and 30) clustered into a novel lineage distantly related to the conserved lepidopteran PR clade. Notably, several DpunPRs (*Shen et al., 2020*) and SlitOR5 (*Bastin-Héline et al., 2019*) that grouped with HrhoOR14 and 30 were recently characterized as a new evolution origin of PRs in Lepidoptera. We speculate that these two ORs may be involved in the detection of *H. rhodope* sex pheromones, although further functional analysis is needed to confirm this hypothesis. Besides phylogenetic analysis, the expression pattern is another main criterion used to select candidate PRs for functional studies. SlitOR5 and DpunPRs were all highly or specifically expressed in the MA. Unfortunately, we did not conduct the expression levels of HrhoOR14 and 30 by qRT-PCR. However, according to the FPKM values, HrhoOR14 exhibited male-biased expression while HrhoOR30 exhibited female-biased expression. This expression pattern was similar to the novel lineage of PRs, whereby approximately one-half of the ORs were male-biased and the other half were female-biased (*Bastin-Héline et al., 2019*).

The expression patterns of candidate genes may reveal important clues into their functions. Overall, sexually differential expressions of ORs in antennae suggest a possible involvement of PRs in sexual behaviors. Female-biased ORs are believed to play a role in detecting oviposition-related odorant (*Pelletier et al., 2010*) or pheromones (signals) released by males (*Anderson et al., 2009*), whereas male-biased ORs potentially function in the detection of sex pheromones released by females (*Zhang & Löfstedt, 2013*). ORs expressed evenly between female and male antennae are predicted to take part in general odorant perception (*Yan et al., 2015*). Therefore, we speculate that male-biased expressed ORs (*HrhoOR4* and *13*) are likely to participate in female pheromone perception. On the other hand, female-biased expressed HrhoORs, including *HrhoOR9*, *24*, *26*, *33*, *37* and *40*, may function in the detection of oviposition-related plant odors or male-produced courtship pheromones. The other HrhoORco and HrhoOR11 with approximately equal expression levels between female and male antennae are likely to function in the detection of food source odors. In addition, HrhoOR37 was orthologous to CmedOR32 according to phylogenetic analysis, and the expression levels of these two OR genes were higher in the FA than in the MA (*Liu et al., 2017b*). Therefore, we speculate that these two orthologous OR genes may play similar roles in odor detection. Detailed functional studies of these sex-specific ORs should be performed in further studies.

IRs are a new subfamily of chemosensory receptors. IRs are relatively conserved in both the sequence and the expression patterns and are widely distributed throughout the body parts, including the labellum, pharynx, leg and wing compared with ORs (*Koh et al., 2014*; *Rimal & Lee, 2018*). In this study, 9 IRs were identified. IR8a, IR25a and IR76b were regarded as co-receptors just like ORco as they were co-expressed along with other stimulus-specific IRs (*Abuin et al., 2011*). HrhoIR8a and HrhoIR76b clustered in the IR8a/IR25a/IR76b subfamily, indicating that these two genes may perform a function similar to that of the co-receptors in *D. melanogaster*. Interestingly, IR25a, one of the conserved IR co-receptors, was not identified in this study. The failure to identify IR25a could be due to the low expression levels in the antennae in this species. Other IRs found in antennae belong to the conserved ''antennal IRs'' (*Croset et al., 2010*) and have their respective orthologs. This revealed that IRs are highly conserved across insect orders and HrhoIRs may have conserved functions, retaining their roles as IRs in other Lepidoptera species. However, the function of these IRs has been studied in *D. melanogaster*. For example, DmelIR21a and 25a are essential for cool sensing (*Ni et al., 2016*), IR40 and IR68a are involved in the detection of temperature and humidity (*Enjin et al., 2016*; *Frank et al., 2017*), IR75 functions in acid sensing (*Prieto-Godino et al., 2017*) and IR76b is involved in the sensation of taste (*Ganguly et al., 2017*). Therefore, HrhoIR21a, HrhoIR40a, HrhoIR68a, HrhoIR75 and HrhoIR76b are predicted to perform functions similar to those in *D. melanogaster*, which may involve activation by acids, temperatures, humidity and other factors. The function of IRs in lepidopteran species is still unclear, warranting further functional studies. In the expression patterns of the 9 *IRs* identified, 8 *HrhoIRs* were highly expressed in the antennae, indicating that these IRs may participate in odor, thermo- and hygrosensation. Three *IRs*, *HrhoIR40a*, *HrhoIR75p* and *HrhoIR75q.2*, were also expressed in the heads and legs except for the antennae. Notably, *HrhoIR75q.2* had the highest expression level in the heads compared with the other four tissues studied. The expression of these three *IRs* in different tissues suggests that they may have multiple functions. In addition to olfaction, IRs are associated with gustation, hygrosensation and thermosensation (*Rimal & Lee, 2018*). In *D. melanogaster*, *IR21a*, *IR40a*, *IR68a* and *IR93a* were expressed in the antennae and played critical roles in thermosensation and hygrosensation (*Enjin et al., 2016*; *Knecht et al., 2016*; *Ni et al., 2016*; *Frank et al., 2017*; *Rimal & Lee, 2018*). In our study, HrhoIR21a and HrhoIR40a showed a close evolutionary relationship to DmelIR21a and DmelIR40a. Additionally, *HrhoIR21a* and *HrhoIR40a* showed high expression levels in the antennae (Figs. 6L–6M). However, further analysis should be carried out to establish whether these two IRs function in the mediation of thermotransduction in *H. rhodope*.

GRs function in detecting $CO_2$ and nonvolatile bitter, sugar, amino acid and plant secondary metabolite compounds via contact chemosensation (*Agnihotri, Roy & Joshi, 2016*). Phylogenetic tree analysis showed that HhroGR2 clustered with BmorGR2 in the $CO_2$ receptor subfamily (Fig. 4). Moreover, just like *BmorGR2*, *HrhoGR2* exhibits higher expression levels in the antennae compared with other tissues (*Guo et al., 2017*). Therefore, HrhoGR2 may contribute to $CO_2$ detection. Furthermore, HrhoGR67 was clustered into a bitter receptor family and highly expressed in the heads (Figs. 4 and 6V). *Kasubuchi et al. (2018)* reported three bitter receptors (*BmorGR16/18/53*) expressed at different levels in

the labrum, maxillary palp and maxillary galea, which perceived various feeding deterrents such as coumarin and caffeine. Further studies should explore whether HrhoGR67 has a similar function to the three receptors.

Generally, there are two SNMPs in insects, which were also identified in the present study. *HrhoSNMP1* and *HrhoSNMP2* were highly expressed in *H. rhodope* antennae, suggesting that they might be involved in olfactory functions. However, *HrhoSNMP2* was also expressed in the heads. Similar broad expression patterns were also observed in other moth species, such as *Sesamia inferens*, *C. medinalis*, *Spodoptera litura* and *H. armigera* (*Zhang et al., 2013*; *Zhang et al., 2015*; *Zhang et al., 2020*). The ubiquitous expression pattern of *SNMP2* means that in addition to odorant detection, they may have various functions specific to different organs (*Vogt et al., 2009*). On the other hand, SNMP1 subfamilies are implicated in mediating responses to lipid pheromones (*Li et al., 2014*). *HrhoSNMP1* showed higher expression levels in the FA, indicating that *HrhoSNMP1* may be associated with the detection of sex pheromones. However, SNMP1 identified from *M. destructor* was reported not to be crucial in pheromone perception (*Andersson et al., 2016*). This finding calls for further investigation to determine whether HrhoSNMP1 functions in pheromone detection in *H. rhodope*.

## CONCLUSION

In conclusion, 45 ORs, 9 IRs, 8 GRs and 2 SNMPs were identified in antennae transcriptomes of *H. rhodope*. The putative functions of some genes were predicted by comparative phylogenetic analyses and tissue expression assays. Our results enrich the olfactory gene inventory of *H. rhodope* and provide the foundation for further research on the molecular mechanism and evolution of the olfactory system in diurnal moths.

## ACKNOWLEDGEMENTS

We thank students Deng-Gao Ji, Jian-Wei Ding and An-Ran Tan (Henan University of Science and Technology) for their help in collecting insects. We also thank Joseph Gillespie, Arthur de Fouchier, Nicolas Montagné and the other anonymous referee for their helpful comments that improved the manuscript.

### Funding

This work was supported by the National Natural Science Foundation of China (31901872), the Ph. D. Programs Foundation of Henan University of Science and Technology (13480048) and the Key Science and Technology Program of Henan Province (182102110255). The funders had no role in study design, data collection and analysis, decision to publish, or preparation of the manuscript.

### Grant Disclosures

The following grant information was disclosed by the authors:

National Natural Science Foundation of China: 31901872.
Ph. D. Programs Foundation of Henan University of Science and Technology: 13480048.
Key Science and Technology Program of Henan Province: 182102110255.

## Competing Interests

The authors declare there are no competing interests.

## Author Contributions

- Haibo Yang conceived and designed the experiments, performed the experiments, analyzed the data, prepared figures and/or tables, authored or reviewed drafts of the paper, and approved the final draft.
- Junfeng Dong performed the experiments, authored or reviewed drafts of the paper, and approved the final draft.
- Ya-Lan Sun and Dingxu Li conceived and designed the experiments, authored or reviewed drafts of the paper, and approved the final draft.
- Zhenjie Hu analyzed the data, prepared figures and/or tables, and approved the final draft.
- Qi-Hui Lyu analyzed the data, authored or reviewed drafts of the paper, and approved the final draft.

## Field Study Permissions

The following information was supplied relating to field study approvals (i.e., approving body and any reference numbers):

We received verbal permission to collect insects in Sui and Tang Dynasties City Ruins Botanical Garden from Yi Zhao the deputy dean of the Garden.

## DNA Deposition

The following information was supplied regarding the deposition of DNA sequences:

All the gene sequences are available at GenBank:

MN515166–MN515221.

## Data Availability

The raw data are available as Supplementary Files.

## Supplemental Information

Supplemental information for this article can be found online at http://dx.doi.org/10.7717/peerj.10035#supplemental-information.

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
