# Peer review of "Identification and expression profiles of candidate chemosensory receptors in Histia rhodope (Lepidoptera: Zygaenidae)"

_PeerJ, doi:10.7717/peerj.10035_

## Round 0.1 · original submission · Major Revisions

Dear Drs. Yang and colleagues:

Thanks for submitting your manuscript to PeerJ. I have now received two independent reviews of your work, and as you will see, the reviewers raised some concerns about the research. Despite this, these reviewers are optimistic about your work and the potential impact it will lend to research on chemosensory receptors in Lepidoptera. Thus, I encourage you to revise your manuscript, accordingly, taking into account all of the concerns raised by the reviewers.

Importantly, please ensure that an English expert has edited your revised manuscript for content and clarity.

Please improve the Introduction and ensure all relevant references are included.

Please improve the content and clarity of the figures and tables and ensure that your hypotheses are clearly outlined.

The reviewers raised many minor concerns about the manuscript. Please address all of these issues.

I look forward to seeing your revision, and thanks again for submitting your work to PeerJ.

Good luck with your revision,

-joe

·

Basic reporting

The manuscript of Yang et al. describes the annotation of candidate chemosensory receptors in an antennal transcriptome that has been previously published by the same authors (in March 2020). In this previous publication, authors only annotated soluble olfactory proteins and here, they describe the identification of chemoreceptors, based on the same RNAseq data. A major concern is that the origin of the transcriptome analyzed here is not well described in the introduction and material & methods (how has this transcriptome been obtained? In which publication is it described?) and raw data availability is not indicated, suggesting that RNAseq data (raw reads) have not been made available.

A short introduction describes the different chemosensory receptor genes studied here, and the biological model. However, the question behind this work is not really clear: authors talk about an evolutionary shift from olfactory to visual communication in diurnal moths such as H. rhodope, but there is no literature cited in the introduction to support this claim. Notably, H. rhodope uses pheromones for sex communication, so the “evolutionary shift” is not obvious. Appropriate references should be included, otherwise this shift should just be presented as a working hypothesis.

Concerning the English language, an additional check is mandatory, as there are a lot of badly written sentences (some are listed in my comments, but NOT ALL). This is especially the case in the discussion section.

Experimental design

As stated above, the research question is not clearly presented in the introduction. The present work is rather descriptive (identification of olfactory genes in an original insect model) so it may have been better to clearly explain that the objective of the study was to describe for the first time the chemoreceptor repertoire of a Zygaenidae species, a group of diurnal moths.

The experimental strategy followed (annotation of sequences in the reference transcriptome, quantification of gene expression using RNAseq, then qRT-PCR) is really classic for this kind of study. However the material & methods section lacks some essential information and some analyses should be redone using different methods. Below you will find several requests and suggestions, by order of importance:

1) For the qRT-PCR experiment, you have to detail the following steps: RNA extraction, RNA purity check and quantification, reverse transcription (kit, reaction conditions), amount of cDNA and reaction volume for the PCR… Please refer to the MIQE guidelines if necessary (Bustin et al 2009). The choice of the 10 OR genes tested also need to be justified, here it seems that they have been picked up randomly. Notably, that’s a pity that the two most interesting OR genes (OR14 and 30) have not been tested.

2) For the qRT-PCR, the choice of GAPDH as a reference gene has not been justified. Please explain how you selected this gene as the best reference, and if this choice has been described previously please cite the reference. This is especially important because you used only one reference gene, whereas it is usually recommended to use at least two.

3) Concerning primer efficiency, it is stated that “the specificity and the efficiency of each primer were validated before” but the methodology should be described (melting curves or gel for specificity? standard curves with 5-fold dilution series of a pool of cDNA for establishing primer efficiency?). Moreover, if you calculated primer efficiencies, then these values should be used for calculating expression levels, rather than just using the 2-ΔΔCt formula.

4) For the search of candidate chemoreceptor sequences, a blastx strategy was used. The blastx results were then searched by keywords. However, it is quite frequent to miss some chemoreceptors by doing so. For example, blast searches with IRs often return sequences named “glutamate receptor” or “Kainate/AMPA receptor” and not necessarily “ionotropic receptor”. For example, it is really surprising that no IR25a sequence is present in the Hrho antennal transcriptome. To ensure that no chemoreceptor sequence is missing in the annotation, this strategy should be completed by a search by domain (InterPro Scan for example) and/or by a tblastn search on the transcriptome, using available lepidopteran receptor amino acid sequences as queries.

5) The source of the reference transcriptome and the RNAseq reads used here is not clearly explained. From what I understand it has already been described in detail in another paper, but please include a minimum of detail (which tissues, how many repeats per sample, how many reads obtained, how many contigs in the reference transcriptome).

6) The tools used for aligning reads and calculating FPKM values are not indicated.

7) For the phylogeny, you did not describe how sequences have been aligned.

8) The OR phylogeny has not been rooted using Orco as an outgroup, the figure has to be redone.

Validity of the findings

In the OR phylogeny, a major finding is the lack of Hrho ORs belonging to the “classical” moth pheromone receptor clade, whereas two HrhoORs have been found in the “novel” PR lineage. This lack of ORs in the classical pheromone receptor clade should be clearly stated in the results section, and not only in the discussion.

The results of the IR phylogeny could be exploited a bit more, notably by reminding the function associated with each IR clade (IR75 > acid sensing, IR21a > cool sensing, IR40a > dry sensing, etc…).

Results of the qPCR experiment are not described in terms of statistical significance (the terms “Highly expressed” are not really informative). A comparison with FPKM values (is it congruent?) should also be included.

Additional comments

Minor comments:

l.56: two CANDIDATE pheromone receptors (…) a novel PR lineage.
l. 66: add a coma after a diurnal moth
l.70: affects (…) disturbs
l.87-88: the sentence should be rephrased
l.89: why “generally”?
l.90: cation channel
l.94: belong to the ionotropic glutamate receptor family
l. 96: antennal IRs also include IRs involved in thermo and hygrosenstion, not only olfaction
l.100: not only in odorant detection (IR76b is most probably involved in taste)
l.112-116: not necessary to cite all the species one after the other, it’s too long.
l.127: only one non-ditrysian moth, Eriocrania
l.129-134: this need to be rephrased. Moreover, there is no reference supporting a shift from olfactory to visual communication. Which article demonstrates an “olfactory related gene loss in diurnal insects ??”
l. 205-207: please rephrase
l.210-211: rephrase
l. 222: rephrase, I guess that all the IRs identified are not full-length
l.223: IRs do not have signal peptides
l. 224-225: rephrase
l. 235: same remark as above, all GRs are NOT complete
l. 237: what do you mean by “the other 3 chemoreceptor genes”? please rephrase
l. 239: analysis, not calculations
l. 283: separata
l. 305: “Orco genes are widely distributed”: what do you mean?
l. 351: “members of this group“: it’s not clear what you are talking about
l.360: antennae are not only involved in olfaction, but also in other sensory modalities. So, expression in antennae does not necessarily mean that these are olfactory ionotropic receptors.
l.366: in thermosensation and hygrosensation
Fig. 2 legend: it is written Ositrinia instead of Ostrinia

Reviewer 2 ·

Basic reporting

No comments

Experimental design

No comments

Validity of the findings

No comments

Additional comments

This study identified some chemosensory receptors genes of male and female of Histia rhodope Cramer from previously acquired H. rhodope antennal transcriptomic data. The authors used this data to curate olfactory-related genes and examine their sex-specific expression. Overall, this work is convincing: the methods used appear adequate, results seem reliable and the conclusions are supported by the data. This research also provides new genomic resources for this specie. However, some issues should be addressed before the manuscript merits to be published.

1. Please remove the full meaning of these abbreviations (OR, IR, GR, SNPS) on the Materials & Methods part (L115-157). I appreciate that the full meaning of these abbreviations is provided in the introduction and no needs to be provided again.

2. The authors have compared the sequences via Blastx and constructed phylogenetic tree, including the OR gene. Therefore, why the author named the OR genes numbered arbitrarily, while IR, GR and SNMP genes were named based on the highest scoring Blastx match from the NR database (L159-160)?

3. Why selected the GADPH gene as the reference gene? Is there any literature to support/report this? Other tissues of H. rhodope you selected were also use this gene to evaluate? Did the authors set any cDNA dilutions before qRT-PCR? This is necessary to check qPCR Standard Curve Slope to find the efficiency of the primer.

4. Line 296 – “H. rhodope is oligophagous pest and does not require additional olfactory proteins to perceive single host chemicals” Is this true? Please mention any supportive evidence (reference will do).

5. Line 322 – “However, according to FPKM values, HrhoOR14 exhibited male-biased expression while HrhoOR30 exhibited female-biased expression” is there any evidence based on sexual biased proved in the OR14 and 30?

6. Figure 4 and 5 – Suggestion – Phylogenetic tree is very small and hard to read the labeling. Increase the size and make it readable.

7. Figure 7 – Suggestion – If possible, enhance the resolution. It looks blurred.

8. Suggestion – Minor language check may need to correct the tense and spelling mistake in the manuscript.

---

## Round 0.2 · accepted · Accept

Dear Drs. Yang and colleagues:

Thanks for revising your manuscript based on the concerns raised by the reviewers. I now believe that your manuscript is suitable for publication. Congratulations! I look forward to seeing this work in print, and I anticipate it being an important resource for groups studying chemosensory receptors in Lepidoptera. Thanks again for choosing PeerJ to publish such important work.

Best,

-joe

·

Basic reporting

All the comments made in my previous review have been addressed:
-English has been corrected
-the context in the introduction is now cleraly explained
-the origin of the RNAseq data is now clearly described

Experimental design

All my suggestions have been taken in consideration, and the material & methods now describes clearly what has been done

Validity of the findings

Exploitation of the results has been improved, as previously suggested

Additional comments

All the comments and suggestions I made in my previous review have been addressed. I have no other comments and think that this manuscript is now acceptable for publication in PeerJ.